# Multi-Scale LBP Texture Feature Learning Network for Remote Sensing Interpretation of Land Desertification

Wuli Wang [1,*], Yumeng Jiang [1], Ge Wang [1], Fangming Guo [1], Zhongwei Li [1] and Baodi Liu [2]

1 College of Oceanography and Space Informatics, Cina University of Petroleum (East China), Qingdao 266580, China; z20160019@s.upc.edu.cn (Y.J.); z20160024@s.upc.edu.cn (G.W.); guofangming@s.upc.edu.cn (F.G.); lizhongwei@upc.edu.cn (Z.L.)
2 College of Control Science and Engineering, China University of Petroleum (East China), Qingdao 266580, China; liubaodi@upc.edu.cn
* Correspondence: wangwuli@upc.edu.cn

**Abstract:** Land desertification is a major challenge to global sustainable development. Therefore, the timely and accurate monitoring of the land desertification status can provide scientific decision support for desertification control. The existing automatic interpretation methods are affected by factors such as "same spectrum different matter", "different spectrum same object", staggered distribution of desertification areas, and wide ranges of ground objects. We propose an automatic interpretation method for the remote sensing of land desertification that incorporates multi-scale local binary pattern (MSLBP) and spectral features based on the above issues. First, a multi-scale convolutional LBP feature extraction network is designed to obtain the spatial texture features of remote sensing images and fuse them with spectral features to enhance the feature representation capability of the model. Then, considering the continuity of the distribution of the same kind of ground objects in local space, we designed an adaptive median filtering method to process the probability map of the extreme learning machine (ELM) classifier output to improve the classification accuracy. Four typical datasets were developed using GF-1 multispectral imagery with the Horqin Left Wing Rear Banner as the study area. Experimental results on four datasets show that the proposed method solves the problem of ill classification and omission in classifying the remote sensing images of desertification, effectively suppresses the effects of "homospectrum" and "heterospectrum", and significantly improves the accuracy of the remote sensing interpretation of land desertification.

**Keywords:** desertification; land cover classification; extreme learning machine; local binary patterns; Horqin Left Wing Rear Banner



## 1. Introduction

An increase in desertified land driven by anthropogenic climate change has been observed globally [1], and this is likely to have profound ecosystem impacts in semiarid lands [2,3], leading to land degradation, soil quality loss, vegetation cover reduction, dust, and other catastrophic environmental problems. Desert ecosystems are fragile and susceptible to rapid change from climatic and anthropogenic disturbances [1]; recent evidence suggests these changes may already be in progress [4]. Land use/cover (LULC) monitoring plays a vital role in effective environmental management, assessment of natural resources, environmental protection, urban planning [5–12], and sustainable development. In this context, the timely and accurate mapping and monitoring of the situation of desertification are critical for scientists and planners in developing effective strategies to address these threats [1].

Traditional methods of monitoring land desertification require direct observations in the field. Usually, they are not only ineffective, expensive, time consuming, and labor intensive, but are also limited to the local scale [13]. It does not meet the need for rapid extraction and updating information on land desertification. Hence, remote sensing with analysis

techniques is highly recommended for the better management of territory and resources. Remote sensing is a critical tool for monitoring environmental transitions [14] and has been instrumental for mapping LULC change since the first launch of Earth observation satellites in 1972 with Landsat-1 because of its objectivity, cost saving, and repetitive coverage over wide spatial and temporal scales [15–17]. Recently, there has been a growing availability of freely available satellite data products and improved classification techniques [18]. Such developments provide a good environment to explore innovative mechanisms capable of improving the accuracy of LULC products, even under complex and heterogeneous landscapes [19–21].

The goal of an image segmentation algorithm is to divide an image into meaningful separate regions that are homogeneous concerning one or more properties, such as texture, color, or brightness [22], and this goal is usually accomplished by image classification at the pixel level. Image segmentation algorithms have been widely used in remote sensing, such as support vector machines (SVMs), random forests (RFs), and convolutional neural networks (CNNs). Nonparametric machine learning algorithms, such as SVM and RF, are well known for their optimal classification accuracies in land cover classification applications [23,24]. Moreover, CNN, a more recently developed but well-represented deep learning method, allows the rapid and effective analysis and classification of LCLUs and has proven to be a suitable and reliable method for accurate classification in complex scenes [25,26]. Munoz et al. [27] developed a land cover classification model with CNNs and a data fusion framework to analyze the coastal wetland dynamics associated with urbanization, the sea level rise and hurricane impacts in the Mobile Bay watershed since 1984. Jozdani et al. [28] combined machine learning methods with object-based image analysis (OBIA) techniques for urban LCLU classification. The multi-layer perceptron model containing GB/XGB and SVM produced highly accurate classification results, demonstrating the versatility of these machine learning algorithms. Niculescu et al.through shallow machine learning algorithms, used RFs for vegetation monitoring in the Pays de Brest (France), and Niculescu et al. [29] applied the algorithms of rotation forest, canonical correlation forests and random forest (RF) for the classification of the different categories of land cover (vegetation) of the peninsula. Although there is overwhelming evidence that the performance of machine learning and deep learning classifiers varies with landscape conditions, studies acknowledge that the potential to fully utilize remote sensing as a reliable source of LULC is yet to be realized.

Remote sensing images are always used to represent the natural geographical world, and ground objects do have complex properties. First, in a specific spectral band, two different objects may present the same spectral line characteristics, and the same objects may show different spectral line characteristics, which are not conducive to the segmentation of ground objects. More and more researchers tend to use spatial features for segmentation. Furthermore, ground objects are always irregular in shape and have a wide range of scales. However, it is not easy to obtain optimal image-segmentation results according to the different scales of ground objects. The multi-scale segmentation strategy is widely used to handle the difficulty of wide-scale ranges. In the remote sensing interpretation of land desertification, areas of different degrees of desertification are interspersed (such as large areas of moderate desertification and patches of heavy desertification scattered together; woodlands, grasslands, lakes, and ponds of different sizes), which often leads to the misclassification of desertification types in large feature areas and the omission of desertification types in small features when interpreting. We can easily conclude that, due to data uncertainty, ground object complexity and disturbance diversiform, the multi-scale segmentation strategy is more suitable than traditional single scale techniques for interpreting remote sensing images of land desertification [30].

This paper aimed to study the significance of integrating spectral features with spatial features on the accuracy of land desertification remote sensing interpretation based on the ELM classifier. Inspired by the idea of multi-scale feature extraction, a multi-scale LBP texture feature extraction structure was proposed to extract ground objects' multi-scale

spatial features. Then, the multi-scale LBP features were fused with original spectral features effectively to produce a comprehensive description of spectral and spatial texture information. This feature extraction method can address the suboptimal classification accuracy due to the heterogeneity of similar objects and complex spatial correlations in satellite images. Finally, by adding adaptive median filtering to improve the ELM, the aim is to smooth the initial probability map of the classification using spatial information, remove the "salt and pepper noise noise", and optimize semantic interpretation results.

Additionally, the research on mapping land desertification by machine learning algorithms remains insufficient, and there is no open desertification dataset for experiments. Therefore, we take the Horqin Left Wing Rear Banner as the research area and select four typical areas to make datasets to validate the proposed interpretation method. In summary, the contributions of this paper are described as follows:

1. The MSLBP texture feature extraction method is proposed to solve the misclassification and omission problems in the interlaced distribution of desertification regions.
2. The fusion of spectral features and MSLBP features(S-MSLBP) provides more comprehensive features of ground objects, effectively solving the classification errors caused by "same spectrum different matter" and "different spectrum same object".
3. Filter-based ELM (ELMF) is an improvement of the ELM classifier using adaptive median filtering. This post-processing strategy makes full use of the local spatial features of remote sensing images and significantly improves the accuracy of remote sensing interpretation of land desertification.
4. We produced multispectral remote sensing interpretation datasets of land desertification in the Horqin Left Wing Rear Banner, an essential part of the largest sandy area in China. This dataset was used to test our proposed S-MSLBP-ELMF framework for the automatic interpretation of remote sensing images of desertification.

## 2. Materials

### 2.1. Study Area and Data Sources

Figure 1 shows the Horqin Left Wing Rear Banner, located in Inner Mongolia Municipality, China.

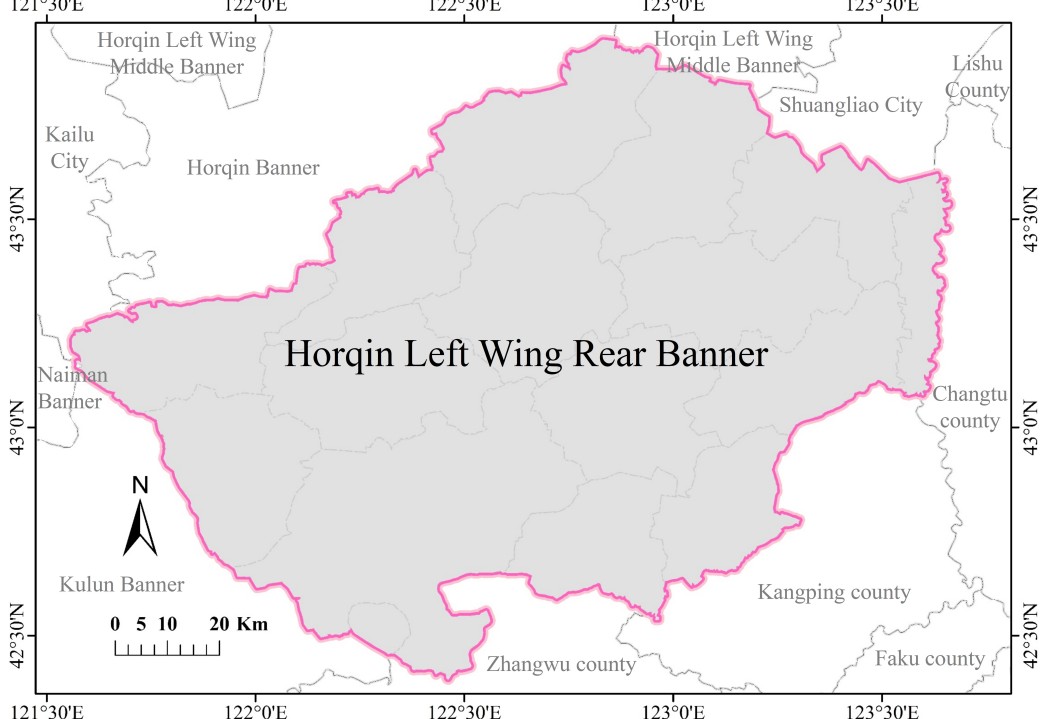

**Figure 1.** The location of Horqin Left Wing Rear Banner.

The Horqin Left Wing Rear Banner was selected as a case study because of its complex surface morphology and the spatial mixture of various land cover types. It is an important part of China's largest sandy area, the Horqin Sands. The primary terrain consists of montane dunes, ribbon valley plains, interdunal depressions, and flat sandy areas. Climatically, the study area is categorized into a subhumid–semiarid climate, which belongs to the marginal zone of the monsoon climate, with four distinct seasons, where irrigated and rainfed drylands are interlaced.

The GF-1 satellite is China's first high-resolution sun-synchronous orbit satellite for Earth observation, with an orbital altitude of 645 km and a return period of 41 days. The satellite carries a 2 m high-resolution panchromatic camera, an 8 m resolution multispectral camera and four 16 m resolution multispectral cameras. This study selected the multispectral remote sensing images available at 16 m spatial resolution from the GF-1 satellite as the experimental dataset. Considering the growth patterns of local crops and small plots of cultivated land in irrigated drylands, we selected multispectral images from 1 September 2020 and images from 14 August as reference data to produce the final label data.

## 2.2. Multispectral Remote Sensing Data Pre-Processing

The acquired remote sensing images are pre-processed on the ENVI remote sensing image processing platform, including radiometric calibration, atmospheric correction, ortho-rectification, and image pruning.

Radiation calibration: The process of converting the digital quantization of an image into a radiometric luminance or reflectance value, here using the Radiometric Calibration tool for radiometric calibration.

Atmospheric correction: Atmospheric correction of multispectral images using the FLAASH atmospheric correction module in ENVI to obtain the spectral properties of features.

Geometric correction and orthorectification: After the usual geometric corrections, the image is then corrected for distortions due to topographic relief according to the DEM and elevation information is added to the image. This step uses the tools in ENVI for geometric correction, orthorectification.

Image cropping: Four typical areas in the back banner of Horqin Left Wing Rear Banner were selected to produce a dataset to facilitate manual visual interpretation and type labeling.

## 2.3. Field Surveys

Land cover data were the basis of this study. In July 2021, we conducted a field survey on the land types of Horqin Left Wing Rear Banner by combining route and point observations to obtain more accurate land cover types in the image area. The field exploration sites are shown in Figure 2.

Based on normalized differential vegetation index (NDVI) calculations and field research, the land cover of the study area was classified into seven categories, including non-desertification, mild wind-eroded desertification, moderate wind-eroded desertification, severe wind-eroded desertification, lakes, mild salinization, and severe salinization.

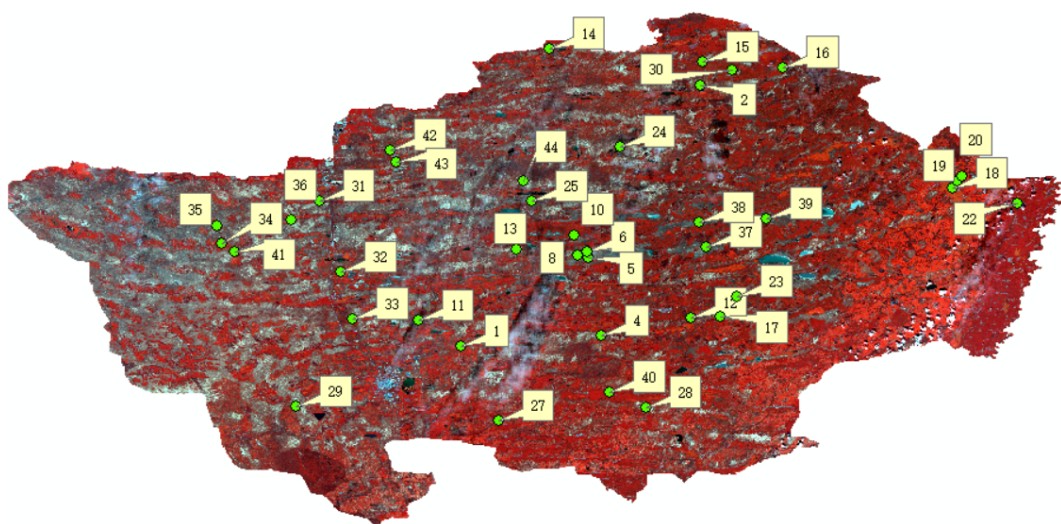

**Figure 2.** Site survey distribution point map in Horqin Left Wing Rear Banner.

*2.4. Datasets Description*

We selected four typical areas to produce the datasets, including saline land, wind-eroded desertification areas, ecological restoration demonstration areas, and complex terrain areas. Figure 3 shows false-color images of the four datasets. Each image is 401 × 401 in size and includes four bands, blue, green, red, and near-infrared. Figure 4 shows the ground truth map of the four datasets, and Table 1 lists the sample sizes for each land cover type in the four datasets.

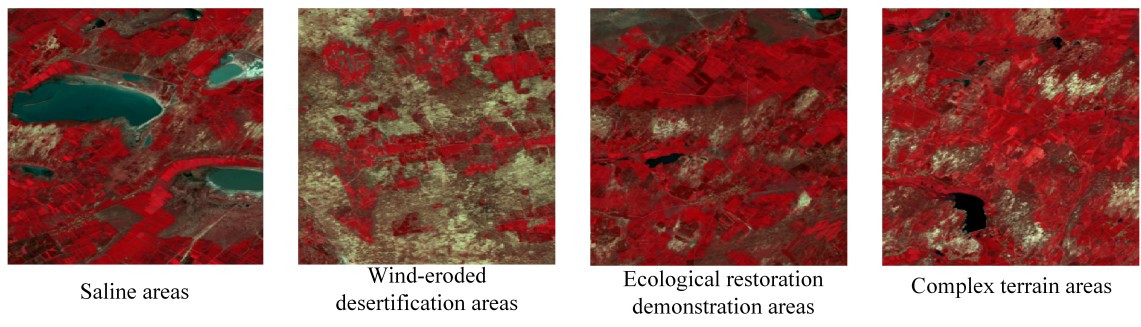

**Figure 3.** False-color images of the four datasets.

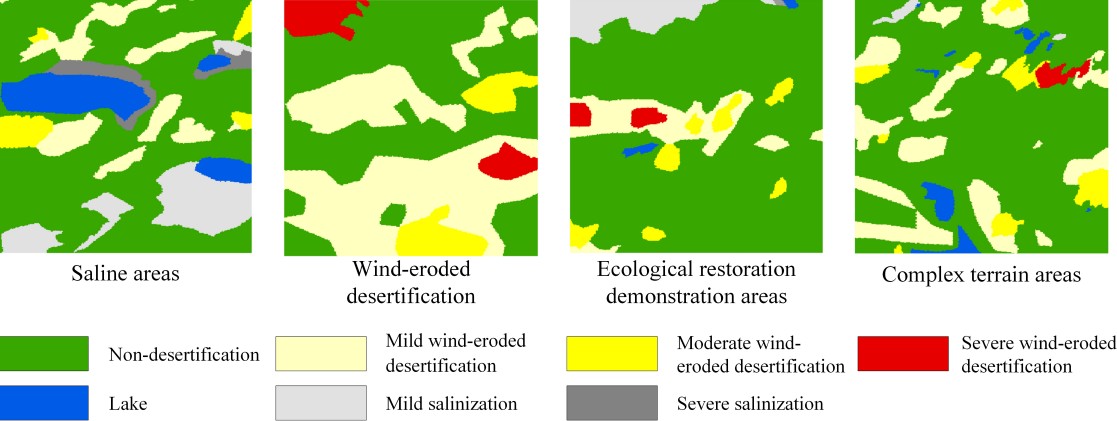

**Figure 4.** Truth map of four datasets.

**Table 1.** Number of samples of each type in four datasets.

| No. | Classes | Saline Areas | Wind-Eroded Desertification | Ecological Restoration Areas | Complex Terrain Areas |
|-----|---------|--------------|-----------------------------|------------------------------|------------------------|
| 1 | Non-desertification | 95,059 | 77,843 | 127,740 | 116,448 |
| 2 | Mild wind-eroded desertification | 18,090 | 61,722 | 13,273 | 28,340 |
| 3 | Moderate wind-eroded desertification | 5878 | 11,713 | 6669 | 8534 |
| 4 | Severe wind-eroded desertification | 0 | 9523 | 2582 | 1957 |
| 5 | Lake | 16,285 | 0 | 871 | 4427 |
| 6 | Mild salinization | 19,465 | 0 | 9528 | 1095 |
| 7 | Severe salinization | 6024 | 0 | 138 | 0 |
| | Total | | | 160,801 | |

## 3. Methodology

In this section, firstly, we design a multi-scale LBP texture feature learning network by analyzing the multi-scale texture features of remotely sensed images. Then, we improve the ELM classifier by adding adaptive median filtering to exploit the local spatial information of the remotely sensed images and improve the remote sensing interpretation accuracy.

### 3.1. Multi-Scale LBP Texture Feature Extraction

3.1.1. Principle of LBP Texture Feature Extraction

Because the remote sensing image frequently appears as the "same spectrum different matter" and "different spectrum same object" phenomenon, it only depends upon the spectrum characteristic and is often insufficient to withdraw the goal object accurately.Texture features can accurately represent the spatial structure information in remote sensing images. The classification method using texture features can effectively suppress the negative impacts of spectral features. LBP is a simple yet efficient advanced operator to describe the local spatial pattern [31]. Due to the advantages of its rotational invariance and low influence by changes in light, LBP is widely used in the remote sensing community.

The original LBP algorithm flow is shown in Figure 5. The LBP operator is defined as in $3 \times 3$ window. Given a center pixel $I_c$ (scalar value), each neighbor of a local region is assigned with a binary label, which can be either "0" or "1", depending on whether the center pixel has an enormous intensity value or not. In this way, there are 8 adjacent pixels around the central pixel, and each pixel is assigned with a binary label. The LBP code is then calculated in a clockwise direction, resulting in an 8-bit binary number, i.e., the binary label sequence of Figure 5 is "01111100", which is converted to a decimal number 127. From this, we obtain the LBP value of the center pixel, which describes the texture structure information between the central pixel and the neighborhood. There are 256 combinations of the 8-bit binary label sequence, so $LBP_8$ can form 256 different binary label sequences.

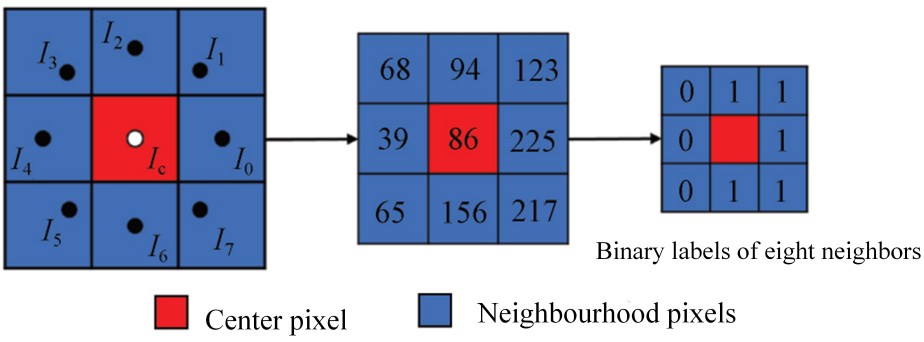

Binary labels of eight neighbors

■ Center pixel     ■ Neighbourhood pixels

**Figure 5.** Example of LBP binary thresholding.

When extracting texture features from remote sensing images, LBP acts on each band of the image, with each band producing 256 dimensional features because there are 256 binary patterns. High-dimensional features will lead to a sharp increase in the computational effort and seriously affect the effectiveness of the model application. The uniform LBP only counts the case where the number of $0-1$ jumps in LBP coding is less than or equal to 2. For example, "01111101" has 3 jumps, so it is not counted. According to the statistics, there are 2 binary label sequences with 0 jumps, 14 binary label sequences with 1 jump and 42 binary label sequences with 2 jumps in the $3 \times 3$ window. More than two jumps are actually rare, so all binary label sequences with more than two jumps are classified as mixed modes. In this way, there are only 59 binary modes to consider. The uniform LBP can effectively extract key regions that describe texture features, such as image edges, blobs, and corners, while significantly reducing the feature dimension. Therefore, using an equivalent model to extract texture features from multispectral remote sensing images can significantly improve the speed of model operations.

### 3.1.2. Multi-Scale LBP Feature Testing and Analysis

Although the equivalent model of LBP can effectively extract texture features, the weak representation ability of single-scale texture features cannot take into account the structural information and detailed features of land desertification types in remote sensing images. This paper expands the field of action of the original LBP operator by convolution, thus obtaining LBP features at more scales. This multi-scale LBP feature extraction method can capture more information about the image structure. Appropriate convolution kernels can effectively remove noise in images and highlight image features. Convolution is the most basic but useful operation in image processing and has two very key features: linearity and translational invariance. Linearity means replacing each pixel with a linear combination of its neighbors, and translation invariance means performing the same operation at every position in the image. Therefore, in this paper, the size of the region block is compressed by convolution operation first, and then LBP feature extraction is performed to obtain larger regional texture features. The principle is shown in Figure 6.

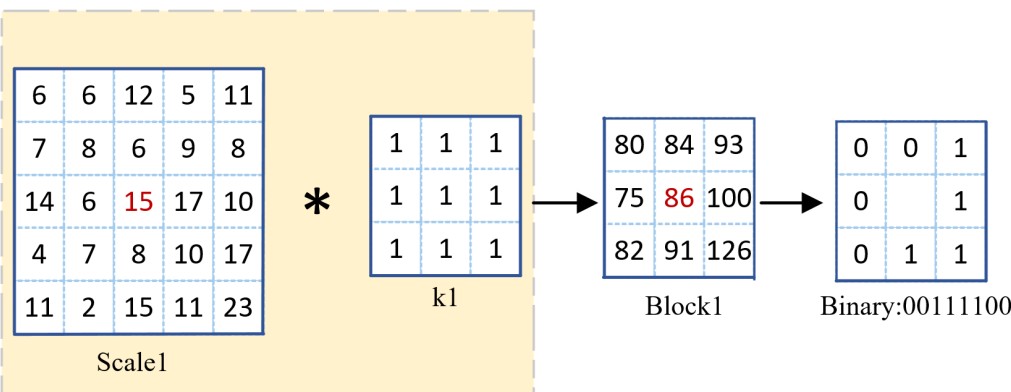

**Figure 6.** Multi-scale LBP feature extraction principle.

Take a $5 \times 5$ block Scale1 centered on a single pixel, convolve it with a convolution kernel k1 of size $3 \times 3$, project the pixels in Scale1 into Block1, and execute the operation shown in Figure 5 on Block1. By combining blocks of different scales with convolution kernels, LBP features under different perceptual fields can be obtained.

This paper verifies through experimental tests that multi-scale LBP texture features are more effective in automatic land desertification interpretation based on remotely sensed images. Figure 7 shows the feature map of arable land in remote sensing imagery with different scale convolution kernels. As shown in the pictures, at the $3 \times 3$ convolution scale, the detailed information about the arable land is precise. At the $5 \times 5$ convolution scale, the detailed information of the arable land is blurred, but the spatial structure is highlighted. The detailed information about the cultivated land is completely lost at the

$7 \times 7$ and $9 \times 9$ convolution scales, while the structural information is roughly preserved and almost lost at the $11 \times 11$ convolution scale. The analysis of the test results shows that the features extracted at the $3 \times 3, 5 \times 5, 7 \times 7$ and $9 \times 9$ scales are consistent with the arable land pattern, detailed features are evident under small-scale convolution, and features extracted by large-scale convolution better represent structural information.

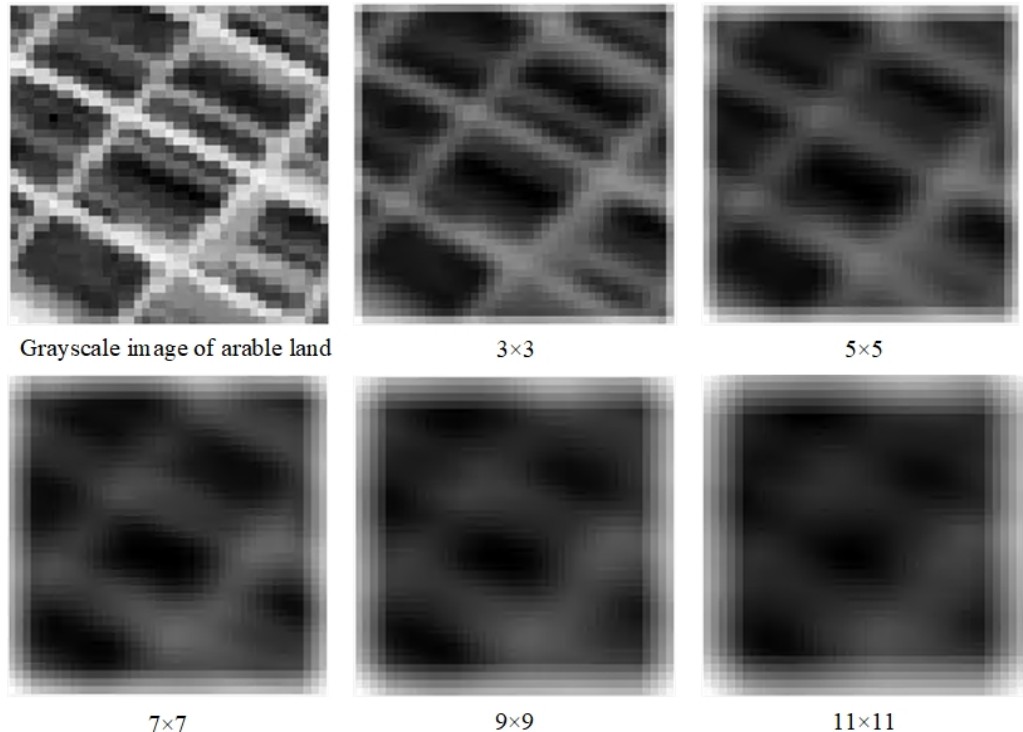

**Figure 7.** Different scale feature map of cultivated land.

To further test the effectiveness of multi-scale LBP features inland desertification classification, we selected typical areas from each of the seven categories of land desertification and extracted the corresponding LBP texture features under different scales of convolution. Figure 8 shows the LBP histograms for the 7 classes of features at different scales.

Significant differences can be seen in the LBP histograms at different scales for the various types of features. The histogram distributions of non-desertification, mild wind-eroded desertification, and moderate wind-erosion desertification at the $5 \times 5$ scale are similar, but there are significant differences in values. The histograms of non-desertification and severe wind-eroded desertification are significantly different in distribution and value. The histogram characteristics of lake and non-desertification at the $9 \times 9$ scale differ considerably and can be distinguished. The test results show that the LBP characteristics of different land types differ significantly at different scales, with a substantial degree of differentiation.

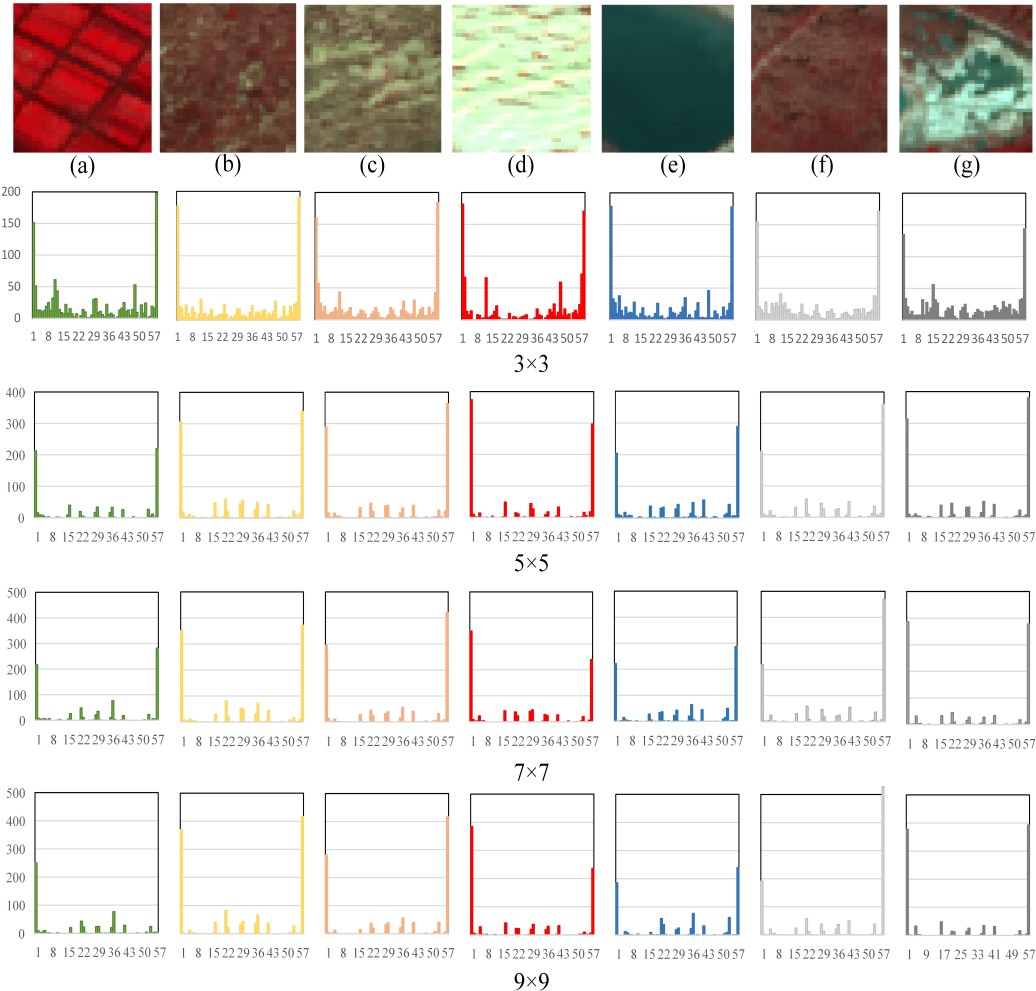

**Figure 8.** Different scale feature map of cultivated land. (**a**) Non-desertification. (**b**) Mild wind-eroded desertification. (**c**) Moderate wind-eroded desertification. (**d**) Severe wind-eroded desertification. (**e**) Lake. (**f**) Mild salinization. (**g**) Severe salinization.

### 3.1.3. Structural Design of Multi-Scale LBP Feature Extraction Network

In summary, it can be seen that the uniform pattern LBP features have fewer dimensions and can effectively extract texture features from remote sensing images. Additionally, the LBP features of remote sensing images of different land types vary significantly at different scales and are highly differentiated. Therefore, we use the uniform pattern LBP as the basis and fuse multi-scale convolutional features to characterize the arrangement and combination of texture primitives inside remote sensing images from a multi-scale perspective and enhance the representation of structural texture information remote sensing images. Therefore, we use the equivalence model LBP as the basis and fuse multi-scale convolution features to characterize the arrangement and combination of texture primitives inside remote sensing images from a multi-scale perspective, to improve the representation capability of structural texture information of remote sensing images and solve the problem of misclassification and under-classification in the interlaced distribution of desertification areas.

The multi-scale convolutional fusion LBP feature extraction network structure is shown in Figure 9. For each band of the remotely sensed image to be processed, firstly, the neighborhood blocks Scale0, Scale1, Scale2, and Scale3 at the scales of $3 \times 3$, $5 \times 5$, $7 \times 7$ and $9 \times 9$ are selected with the target pixel as the center. Secondly, the LBP features are extracted directly for the neighborhood block Scale0, while for Scale1, Scale2, and Scale3, the features need to be downsampled before extraction. Specifically, the convolution of Scale1, Scale2, and Scale3 with convolution kernels K1, K2, and K3 are of sizes $3 \times 3$, $5 \times 5$

and $7 \times 7$ respectively so that the local spatial information is compressed into $3 \times 3$ blocks Block1, Block2, and Block3, and then LBP features are extracted. Finally, the LBP features from different scales are fused to obtain multi-scale LBP features.

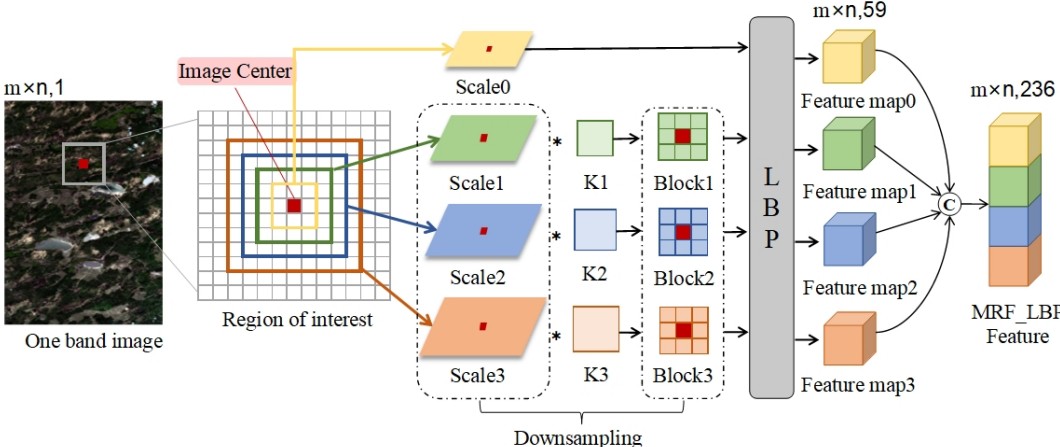

**Figure 9.** MSLBP structures.

### 3.2. Spatial-Spectral Feature Fusion

Spectral features are widely used for remote sensing image classification [32,33]. Due to the influence of the surrounding environment, the relative angle of the sunlight, or the growth environment, there is the phenomenon of "same spectrum different matter" and "different spectrum same object" which leads to the problem of misclassification in the interpretation of remote sensing image. However, these are only a few cases; the contribution of spectral features to ground object classification cannot be ignored. Different ground objects have different spectral characteristics depending on their microstructure and macroscopic properties, which are ground objects' general characteristics. Despite some bias, the spectral features are still valuable.

The integration of rich spectral features information and the extracted multi-scale LBP spatial texture features can better represent the characteristics of various types of features desertification and is more conducive to distinguishing different feature types. This fusion of spatial–spectral features solves the problem of "same spectrum, different objects, different spectra" and effectively improves the accuracy of remote sensing interpretation of land desertification. The classification frame of the fusion of spatial-spectral features is shown in Figure 10.

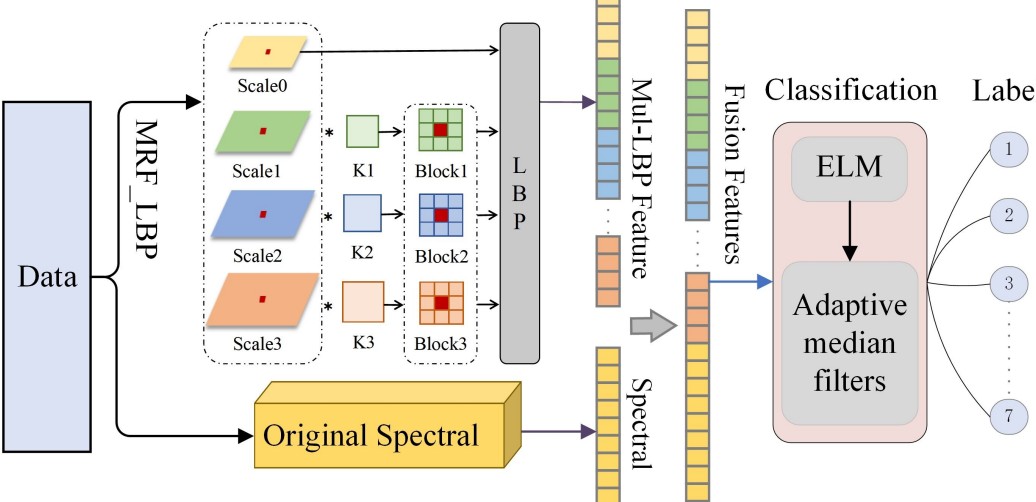

**Figure 10.** The classification frame of the fusion of spatial-spectral features.

### 3.3. Classifier Design

ELM [34,35] is a neural network with only one hidden layer and one linear output layer. The weights between the input and the hidden layers are randomly assigned, and the weights of the output layer are computed using a least-squares method. Therefore, the computational cost is much lower than any other neural network-based method. In addition, ELM has been successfully applied to remote sensing images, biosciences, and Earth sciences with a faster learning rate and better generalization than SVM [36,37]. Given the enormous volume of data in the automatic interpretation of remote sensing images of land desertification, overfitting may occur during training. We have chosen adaptive ELM [38], which can continuously change the number of nodes in the hidden layer during training, balancing empirical and structural risks in a way that avoids overfitting.

Similar features may show local deviations on remote sensing images due to sensors and environmental factors. The diversity of feature texture features and the similarity of spectral features can lead to the misclassification of classifiers and reduce the accuracy of land desertification interpretation. According to the first law of geography (Tobler's First Law of Geography: everything is related to everything else, but near things are more related to each other), we believe that similar features in remote sensing images of land desertification will be clustered and distributed, and the image elements scattered within a certain distance will show spatial dependence in numerical characteristics. The filtering method can maintain the spatial contextual relationship of remote sensing images very well. In addition, the variety and size of features in the remote sensing interpretation of land desertification make a single-scale filtering window impossible for balancing denoising effects and detail maintenance. Therefore, we design an adaptive median filter to post-process the initial probability map output of the ELM classifier to improve the accuracy of the remote sensing interpretation of land desertification. The structure is shown in the second half of Figure 10.

Let $S_{xy}$ denote the filter window size; $x$ and $y$ represent the number of rows and columns of pixels in the window, respectively; $S_{max}$ represent the maximum preset window size; $S_{min}$ represent the minimum preset window size; $Z_{min}$, $Z_{max}$, and $Z_{med}$ be the median of the minimum probability, maximum probability and probability in $S_{xy}$; and $Z_{xy}$ be the probability value of the pixel point in the xth row and yth column of the image. $X \in \mathbb{P}^{m \times n \times c}$ is the probability map of the ELM classifier output; $m$ and $n$ denote the image length and width, respectively; and $c$ is the number of categories.

The adaptive filtering process for each type of probabilistic input $X \in \mathbb{P}^{m \times n \times c}$ is as follows:

Step 1: Determine the noise sensitivity of the current area $S_{xy}$, If the condition of $Z_{min} < Z_{med} < Z_{max}$ is not satisfied, increase $S_{xy}(S_{min} \leq S_{xy} \leq S_{max})$ and repeat the judgment until the condition is satisfied.

Step 2: Determine whether the central pixel probability value $Z_{xy}$ satisfies the condition of $Z_{min} < Z_{xy} < Z_{max}$, and if so, keep the original value; otherwise $Z_{xy} = Z_{med}$.

Step 3: Traverse the entire probability map until processing is complete.

The improved ELM classifier based on adaptive median filtering greatly reduces the misclassification points generated by the automatic interpretation of remote sensing images of land desertification and significantly improves the interpretation accuracy of land desertification.

### 3.4. Parameter Tuning

There are two aspects of the parameters to be considered: the training samples and the adaptive median filter. In this section, we employ control variables experiments with S-MSLBP-ELM as the baseline method to explore the effect of the window size of the adaptive median filter and the proportion of training samples on the classification performance.

The window size of the adaptive median filter is determined by $S_{min}$ and $S_{max}$. The study found that $S_{min}$ affects the classification accuracy, so it is necessary to explore to explore the appropriate value of $S_{min}$. The value of $S_{max}$ is fixed at 25. As shown in Table 2, when the value of $S_{min}$ is 11 or 13, the classification accuracy is the highest.

**Table 2.** Interpretation accuracies of different $S_{min}$ in saline areas dataset (%).

| $S_{min}$ / Class | 3 | 5 | 7 | 9 | 11 | 13 | 15 |
|---|---|---|---|---|---|---|---|
| 1 | 99.47 | 99.84 | 99.90 | 99.91 | 99.92 | 99.93 | 99.92 |
| 2 | 99.66 | 99.94 | 100.00 | 100.00 | 100.00 | 100.00 | 100.00 |
| 3 | 99.19 | 99.51 | 99.59 | 99.59 | 99.59 | 99.49 | 99.46 |
| 5 | 100.00 | 99.98 | 99.98 | 100.00 | 100.00 | 100.00 | 100.00 |
| 6 | 99.85 | 99.95 | 99.98 | 100.00 | 100.00 | 100.00 | 100.00 |
| 7 | 99.97 | 100.00 | 99.97 | 100.00 | 100.00 | 100.00 | 100.00 |
| OA | 99.60 | 99.87 | 99.92 | 99.93 | 99.94 | 99.94 | 99.93 |
| kappa | 99.34 | 99.79 | 99.87 | 99.89 | 99.90 | 99.90 | 99.89 |

Training samples are critical for machine learning. As shown in the Table 3, the classification accuracy improves with the increase in the sample size. When the proportion of training samples is 10%, the classification accuracy reaches saturation.

**Table 3.** Interpretation accuracies of different training samples in saline areas dataset (%).

| Sample Size / Class | 1% | 3% | 5% | 7% | 10% | 15% |
|---|---|---|---|---|---|---|
| 1 | 98.20 | 99.87 | 99.86 | 99.90 | 99.91 | 99.91 |
| 2 | 90.13 | 99.84 | 100.00 | 100.00 | 100.00 | 100.00 |
| 3 | 98.20 | 99.93 | 99.75 | 99.59 | 99.59 | 99.54 |
| 5 | 99.55 | 99.99 | 99.96 | 99.99 | 100.00 | 100.00 |
| 6 | 96.69 | 99.87 | 99.99 | 100.00 | 100.00 | 100.00 |
| 7 | 98.26 | 99.65 | 99.67 | 99.77 | 100.00 | 100.00 |
| OA | 97.17 | 99.87 | 99.89 | 99.92 | 99.93 | 99.93 |
| kappa | 95.35 | 99.79 | 99.82 | 99.86 | 99.89 | 99.88 |

## 4. Results

We used the four datasets produced above, namely saline areas dataset, wind-eroded desertification dataset, ecological restoration areas dataset, and complex terrain areas dataset, to conduct experimental tests to verify our proposed automatic land desertification interpretation algorithm. All four datasets are of size $401 \times 401 \times 4$. The saline areas dataset contains 6 land cover types; the wind-eroded desertification dataset contains 4 land cover types; the ecological restoration areas dataset contains 7 land cover types; and the complex terrain areas dataset contains 6 land cover types.

The overall accuracy (OA) and kappa coefficient are used to quantitatively evaluate the performance of remote sensing interpretation of land desertification. The OA describes the proportion of correctly classified samples to the total number of samples. The kappa coefficient is based on the confusion matrix to calculate the consistency of the sample and reflects the accuracy of the classification. In the performance of the proposed S-MSLBP-ELMF, we conducted an ablation experiment with different features in four experimental data under the condition of 10% training samples as shown in Tables 4–7.

To verify the effectiveness of the designed MSLBP texture features and ELMF in remote sensing interpretation of land desertification combinations, the classification results of various combinations of features, such as spectral features (S-ELM), LBP features (LBP-ELM), spectral features fused with LBP features (S-LBP-ELM), and spectral features fused with multi-scale LBP features (S-MSLBP-ELM), based on the ELM classifier are also given in the table.

**Table 4.** Interpretation accuracies of different methods on saline areas dataset (%).

| Class \ Method | S-ELM | LBP-ELM | S-LBP-ELM | S-MSLBP-ELM | S-MSLBP-ELMF |
|---|---|---|---|---|---|
| 1 | 81.08 | 96.81 | 97.19 | 97.62 | 99.92 |
| 2 | 43.69 | 93.37 | 93.45 | 94.55 | 100.00 |
| 3 | 66.32 | 93.37 | 93.45 | 94.55 | 99.59 |
| 5 | 91.63 | 98.03 | 98.03 | 98.16 | 100.00 |
| 6 | 53.41 | 95.99 | 96.30 | 96.74 | 100.00 |
| 7 | 54.46 | 93.16 | 93.97 | 93.94 | 100.00 |
| OA | 75.33 | 96.26 | 96.56 | 97.02 | 99.94 |
| kappa | 57.32 | 93.85 | 94.35 | 95.12 | 99.90 |

**Table 5.** Interpretation accuracies of different methods on wind-eroded desertification dataset (%).

| Class \ Method | S-ELM | LBP-ELM | S-LBP-ELM | S-MSLBP-ELM | S-MSLBP-ELMF |
|---|---|---|---|---|---|
| 1 | 72.52 | 97.75 | 97.77 | 98.11 | 99.90 |
| 2 | 55.77 | 97.29 | 97.28 | 97.59 | 99.82 |
| 3 | 46.04 | 97.73 | 97.63 | 98.02 | 99.94 |
| 4 | 36.65 | 97.30 | 97.34 | 97.64 | 100.00 |
| OA | 63.79 | 97.52 | 97.55 | 97.88 | 99.91 |
| kappa | 37.65 | 95.55 | 95.97 | 96.52 | 99.80 |

**Table 6.** Interpretation accuracies of different methods on ecological restoration areas dataset (%).

| Class \ Method | S-ELM | LBP-ELM | S-LBP-ELM | S-MSLBP-ELM | S-MSLBP-ELMF |
|---|---|---|---|---|---|
| 1 | 86.88 | 98.81 | 98.82 | 99.07 | 99.92 |
| 2 | 34.84 | 93.38 | 93.59 | 94.37 | 99.50 |
| 3 | 51.33 | 93.04 | 93.97 | 95.03 | 99.94 |
| 4 | 51.75 | 96.47 | 96.68 | 97.27 | 100.00 |
| 5 | 84.52 | 90.44 | 94.12 | 90.08 | 98.08 |
| 6 | 81.55 | 98.16 | 98.23 | 98.41 | 98.88 |
| 7 | 83.33 | 85.94 | 86.478 | 88.46 | 100.00 |
| OA | 84.64 | 98.01 | 98.10 | 98.40 | 99.82 |
| kappa | 46.69 | 94.36 | 94.63 | 95.49 | 99.48 |

**Table 7.** Interpretation accuracies of different methods on dataset with complex terrain areas dateset (%).

| Class \ Method | S-ELM | LBP-ELM | S-LBP-ELM | S-MSLBP-ELM | S-MSLBP-ELMF |
|---|---|---|---|---|---|
| 1 | 84.86 | 96.95 | 97.12 | 97.55 | 99.56 |
| 2 | 51.76 | 92.80 | 92.66 | 93.79 | 99.85 |
| 3 | 50.90 | 94.20 | 95.55 | 95.76 | 99.88 |
| 4 | 44.03 | 85.91 | 86.43 | 87.39 | 100.00 |
| 5 | 84.24 | 90.22 | 90.51 | 91.83 | 100.00 |
| 6 | 54.47 | 87.72 | 88.81 | 89.62 | 100.00 |
| OA | 78.74 | 95.71 | 95.90 | 96.46 | 99.65 |
| kappa | 45.81 | 90.19 | 90.63 | 91.93 | 99.20 |

From the average interpretation results on the four datasets, the accuracy of remote sensing interpretation of land desertification based on LBP texture features is much higher than that of the spectral feature-based method, the OA is improved by about 20% and the kappa coefficient is improved by about 40%, further validating the effectiveness of LBP texture features for classification. The S-MSLBP-ELMF model has improved classification accuracy on all four datasets compared to the LBP-ELM model uses only single-scale LBP features. Specifically, there is 3.54% improvement in OA and 5.82% improvement in kappa

coefficient on the saline areas dataset; 2.39% improvement in OA and 4.25% improvement in kappa coefficient on the wind-eroded desertification dataset; 1.81% improvement in OA and 5.12% improvement in kappa coefficient on the ecological restoration areas dataset; and 3.94% improvement in OA and 9.01% improvement in kappa coefficient on the complex terrain areas dataset.

From the above experimental results, the performance of our method is significantly better than that of the spectral-based method and the single-scale LBP-based method. In particular, the results of the interpretation of the feature-scale difference dataset show a significant increase in the kappa coefficient, which validates the effectiveness of our method in solving the misclassification and under-classification problems that exist in the staggered distribution of desertification areas.

In order to visualize the performance of the proposed method for the remote sensing interpretation of land desertification, we present in Figures 11–14 the results of the visual interpretation of the five methods on the four datasets. Clearly, maps generated from a classification using spatial features (LBP or MSLBP) are less noisy and more accurate than those from using spectral features. Moreover, MSLBP-ELMF-based methods yield cleaner and smoother maps than LBP-ELM-based methods. Specifically, the classification map of S-MSLBP-ELMF in Figure 11 is more accurate than the map of LBP-ELM. This is because the remote sensing image itself has noise and cannot be exactly the same as the labeled sample. The experimental results prove that S-MSLBP-ELMF is effective, the results obtained are the closest to the ground truth map, and no obvious classification errors appear.

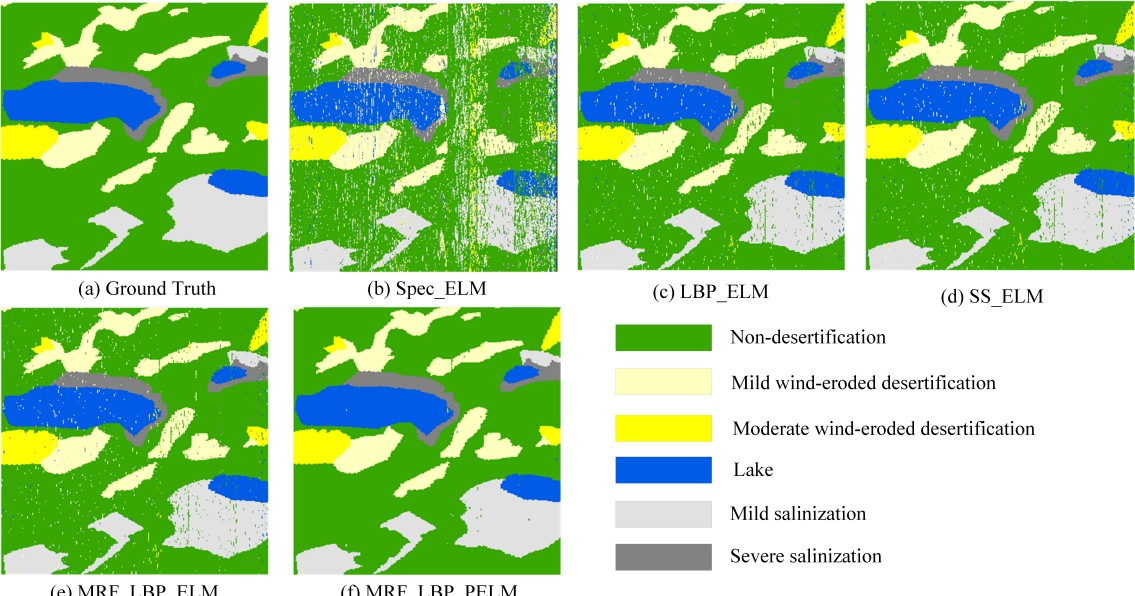

**Figure 11.** Comparison of visual interpretation results of different methods on saline areas datasets.

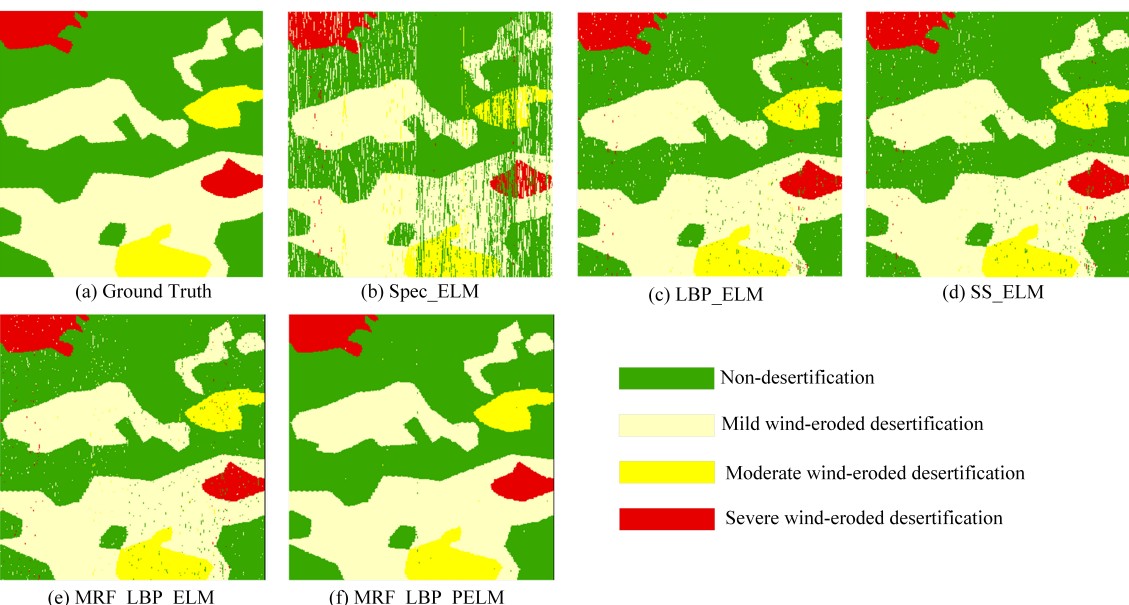

**Figure 12.** Comparison of visual interpretation results of different methods on wind-eroded desertification areas datasets.

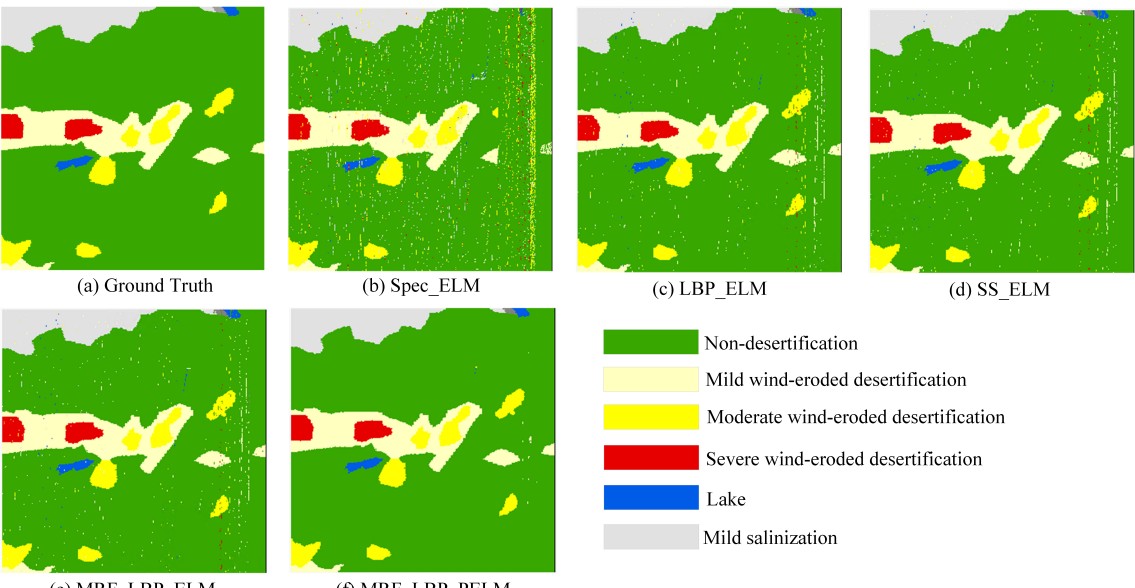

**Figure 13.** Comparison of visual interpretation results of different methods on ecological restoration demonstration areas datasets.

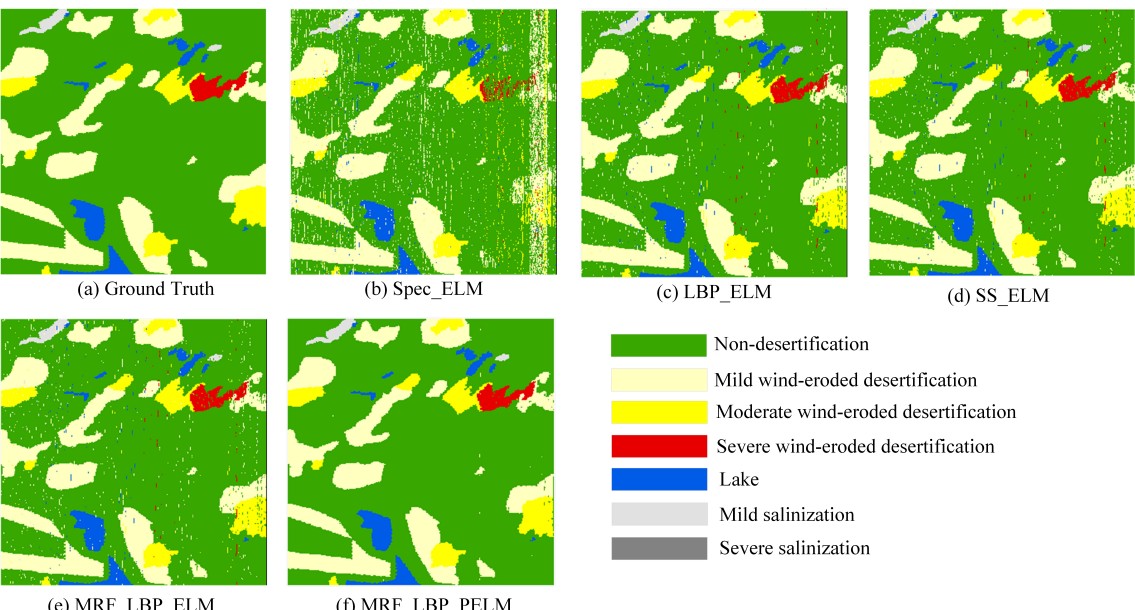

**Figure 14.** Comparison of visual interpretation results of different methods on complex terrain areas datasets.

## 5. Discussion

This study examined whether the remote sensing image segmentation algorithm can provide accurate and reliable results on land desertification interpretation. It has to be stressed that the number of studies on the semi-automatic or automatic interpretation methods of desertification is rather limited. Only a few studies calculate the NDVI using multispectral satellite images to study desertification in specific regions. Most other studies have focused on classifying urban, farmland, and woodland land cover types. Compared to the datasets of other studies for land cover mapping, there is little variation between classes in the land desertification dataset, the criteria for classifying the various types of land cover are not clear, and the boundaries between regions are blurred. In our proposed S-MSLBP-ELMF framework, machine learning needs to rely on sample data to obtain the essential characteristics of the target and to predict and discriminate the unknown data accordingly, so it is important to establish a scientific database of land desertification, which provides the basis for training and prediction of machine learning.

Based on extensive research into desertification information with the NDVI and field-work, the land cover of the study area was classified into seven categories: non-desertification, mild wind-eroded desertification, moderate wind-eroded desertification, severe wind-eroded desertification, lakes, mild salinization, and severe salinization. Then, according to the characteristics and differences of different desertification types on remote sensing images, the interpretation markers of land desertification were constructed through visual interpretation. In labeling the dataset, each land type was judged and distinguished by its size, shape, spatial structure, texture, color, and temporal pattern. Non-desertification includes areas such as woodlands, grasslands, croplands, and settlements, which exhibit characteristic features of remote sensing images. In September, woodland and grassland are in full bloom, so woodland and grassland appear as large reds on the pseudocolor image. The cultivated areas are the product of artificial activities and are large, with significant artificial traces and more regular textures and shapes. The villages have complex spectral characteristics, with light green mixed with red and other color spots on the images, and the villages are surrounded by arable land. NDVI values for the various desertification types and their main reference indicators are shown in Table 8. Horqin Left Wing Rear Banner covers a wide area, and its remote sensing images reach a size of 3828 pixels 7829 pixels. The cost of annotating remotely sensed images of the entire Horqin Left Wing Rear Banner is extremely high, while training and classifying large datasets can lead to a lack of computer

memory. To reduce the resource consumption and better represent the model performance, we selected four typical regions to make datasets for the experiment. Experimentally, our dataset proved to be reliable.

**Table 8.** Categories and features of surface features in study area.

| Classes | NDVI (%) | Reference Indications and Interpretative Signs |
|---|---|---|
| Non-desertification | <70 | Overall red color with over 70% red area |
| Mild wind-eroded desertification | 50–70 | Light red discontinuous distribution |
| Moderate wind-eroded desertification | 10–50 | Red and white spots interspersed |
| Severe wind-eroded desertification | 0–50 | Light yellow or yellowish-white base with red spots |
| Lake | | Dark blue, single tone |
| Mild salinization | 0–30 | Predominantly grey with bright white spots |
| Severe salinization | >30 | Overall bright white |

For remote sensing image segmentation, the extraction of useful features is crucial. It can be seen as a property that reflects the spatial distribution of image pixels and is often characterized by local irregularities and macroscopic patterns. The LBP algorithm is a popular texture feature extraction algorithm that has gained widespread use because of its excellent ability to depict local texture features in images. However, the traditional LBP has a small area of action and is susceptible to noise interference when comparing the size of two adjacent pixels, and the features extracted are relatively homogeneous. We developed a framework named MSLBP to characterize the arrangement of texture primitives within images at multiple scales, better capture the full range of structural features of remotely sensed images and their detailed information, and reveal the unique characteristics of features at different scales of sensory field. As can be found in Tables 4–7, the MSLBP had higher OAs than the LBP since it extracted additional features. However, the misclassified stray pixels, which largely appeared in the classification maps of MRF-LBP-ELM in Figures 11–14, were due to environmental factors that bias the local point data of remote sensing images. These isolated points of classification noise are called "salt and pepper noise", which could be corrected by some post-classification processes.

The initial probabilities obtained from remote sensing image classification are numerical predictions of the real world, with spatial dependence between pixels. Specifically for a certain image region, pixels scattered over a certain distance usually show spatial dependence in terms of numerical characteristics. We introduced an adaptive median filtering approach to the ELM classifier for post-classification processing, as shown in the classification maps, where the improved ELMF classifier removes "salt and pepper noise" while maintaining the image structure, producing a smooth classification map and further improving classification accuracy.

Although our framework achieves impressive results in interpretation accuracy, it more or less has some flaws. First, although within the acceptable limits, multi-scale LBP features require more computational time. Second, adding adaptive median filtering to the classifier can reduce the overall noise level. They may introduce additional errors for some correctly classified pixels because of the aggregation of non-isolated noise. In addition, although noisy labels mislead training to some extent, they still have sufficient spatial and spectral information, which can be beneficial if used properly.

In future research, we will focus on more important features in the MSLBP module and reduce feature redundancy to optimize computing resource overhead. We will try to design a robust remote sensing image classification module that can directly train the noise-robust model on corrupted datasets. We will enhance the generalization capability of the model to be applied to remote sensing images of other desertification areas and combine them with deep learning methods to solve more problems.

## 6. Conclusions

In this paper, GF-1 multispectral data from the Horqin Left Wing Rear Banner in September 2020 were used as the data source to produce a land desertification dataset for four typical regions, and a high-precision remote sensing interpretation network for land desertification incorporating multi-scale LBP texture features and spectral features was also proposed. The network solves the problem of misclassification and under-classification in the interlaced distribution of desertification areas by extraction of multi-scale LBP texture features. The fusion of multi-scale LBP features with spectral features efficiently suppresses the effects of "same-spectrum foreign objects" and "same spectrum different spectrum." The designed adaptive median filter greatly improves the accuracy of the remote sensing interpretation of land desertification. The decoding results on four datasets validate the superiority of the proposed method.

**Author Contributions:** Conceptualization, W.W., Y.J., Z.L., G.W., B.L. and F.G.; methodology, W.W. and Y.J.; software, Y.J.; validation, W.W. and Y.J.; formal analysis, Y.J.; investigation, W.W., Y.J., G.W., F.G. and Z.L.; resources, W.W., Y.J. and Z.L.; data curation, Y.J., G.W. and F.G.; writing—original draft preparation, Y.J.; writing—review and editing, W.W., Y.J., B.L.; visualization, W.W. and Y.J.; supervision, W.W., F.G., B.L. and Z.L.; project administration, W.W., F.G. and Z.L.; funding acquisition, W.W. and Z.L. All authors have read and agreed to the published version of the manuscript.

**Funding:** This research was supported by the Joint Funds of the Fundamental Research Funds for the Central Universities under Grant 27R2117001A, in part by the National Natural Science Foundation of China under Grant U1906217, in part by the Shandong Social Science Planning under Grant 21CSDJ74, and in part by the Fundamental Research Funds for the Central Universities under Grant 22CX01004A-8.

**Data Availability Statement:** All data used and generated in this study are available on request from the author.

**Acknowledgments:** The authors would like to thank Qianqian Wu, Bin Xu, Kaixuan Gong, Ziqi Xin, and Shunxiao Shi of the Horqin Left Wing Rear Banner Land Desertification Project Team for their contribution to desertification research. The author would like to thank Yinshan Bao and Yongfang Wang of Inner Mongolia Normal University for their guidance. The authors would like to thank Guangbo Ren of the First Institute of Oceanography, Ministry of Natural Resources for his help. The authors would like to thank all colleagues in the laboratory for their generous help, especially Wenzong Jiang and Wenhui Guo. The authors would like to thank the anonymous reviewers for their constructive and valuable suggestions.

**Conflicts of Interest:** The authors declare no conflict of interest.

## Abbreviations

The following abbreviations are used in this manuscript:

| | |
|---|---|
| LBP | local binary pattern |
| LULC | land use/cover |
| ELM | extreme learning machine |
| SVM | support vector machines |
| NDVI | normalized difference vegetation index |
| RF | random forests |
| CNN | convolutional neural networks |
| OBIA | object-based image analysis |
| MSLBP | multi-scale LBP |
| OA | overall accuracy |
| S-MSLBP | fusion of spectral features and MSLBP features |
| ELMF | filter-based ELM |
| S-ELM | spectral with ELM |
| LBP-ELM | LBP with ELM |
| S-LBP-ELM | spectral-LBP with ELM |

S-MSLBP-ELM      spectral-MSLBP with ELM
S-MSLBP-ELMF    spectral-MSLBP with ELMF

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
