# Peer review of "Multi-Scale LBP Texture Feature Learning Network for Remote Sensing Interpretation of Land Desertification"

_remotesensing, doi:10.3390/rs14143486_

Round 1
Reviewer 1 Report
This paper presents a model for remote sensing image interpretation of land desertification, which consists of multi-scale local binary pattern (MSLBP) and extreme learning machine (ELM). This task is interesting and important. Experiments show the effectiveness of the proposed method. This work targets on an important issue. Some concerns are as follows:
1. Although a combination of NDVI, field exploration and visual interpretation was used to produce the dataset labels, validation of the reliability of the labels was lacking.
2. The adaptive median filtering approach is introduced to the ELM classifier for post-classification processing. Can it be used in practice?
3. This work uses different works "interpretation", "segmentation" and "classification" to show the same meaning. Why? It is hard to follow.
4. It is strongly recommended that the author adjust the font in Figure 4 to an appropriate size.
5. Too many typos and grammar issues, which should be carefully revised.
6. The symbols and the mathematical presentation are hard to follow.
Reviewer 2 Report
This manuscript proposes an LBP-based approach for land desertification classification based on texture information. The experiments perform well showing promising accuracy. However, I suggest adding some critical details before considering accepting for publication.
1. Suggest adding a map showing the location of the study area
2. Suggest adding the detail of LBP, and uniform LBP. The readers might not be familiar with the model and not know how the 59 bins are calculated.
3. Implementation information is suggested, was the model implemented by frameworks like PyTorch, or from scratch?
4. For CNN and similar models, I believe the discussion of batch size/epochs is critical. Please consider adding more hyperparameter tuning details about the model.
Reviewer 3 Report
The authors propose the multi-scale LBP texture learning network to monitor land desertification. Several comments are given as follows:
1. The multi-scale LBP uses the kernel with 3x3, 5x5, 7x7, 9x9, and 11x11 to illustrate the multi-scales presentations. How to evaluate the relationships between the 'scale' and ground resolutions.
2. The authors proposed the multi-scale LBP and the adaptive median filtering to process the data. In doing so, the low-frequency information will be significantly lost such that the classifications fail. Can the authors provide more explanations.?
3. Usually, there are several ways to complete the multi-scale approach. For example, using the Gaussian function with different window sizes, or using the wavelet transform to decompose the given image into different scales. Why do the authors introduce kernels of different sizes? The authors maybe can provide more descriptions.
The authors did great work and it is supposed to encourage them to submit the revision.
Round 2
Reviewer 1 Report
All previous comments have been well addressed. I have no further comments this time.